# Impacts of Soil and Water Conservation Practice on Soil Moisture in Debre Mewi and Sholit Watersheds, Abbay Basin, Ethiopia

**Bekele Bedada Damtie** [1,2,3,*] **, Daniel Ayalew Mengistu** [1,3]**, Daniel Kassahun Waktola** [4] **and Derege Tsegaye Meshesha** [3,5]

1   Department of Geography and Environmental Studies, Bahir Dar University, Bahir Dar 079, Ethiopia; dan952003@gmail.com
2   Institute of Land Administration, Bahir Dar University, Bahir Dar 079, Ethiopia
3   Geospatial Data & Technology Center, Bahir Dar University, Bahir Dar 079, Ethiopia; derege2014@gmail.com
4   Department of GIS, Austin Community College, Austin, TX 78752, USA; daniel.kassahun@gmail.com
5   College of Agriculture and Environmental Sciences, Bahir Dar University, Bahir Dar 1289, Ethiopia
*   Correspondence: bekelebedada031@gmail.com; Tel.: +251-969016451

**Abstract:** Soil and water conservation (SWC) practices have been widely implemented to reduce surface runoff in the Debre Mewi watershed. However, studies on the issue have disproportionately focused on the lost or preserved soils, expressed in tons per hectare, while the impacts on the lost or preserved moisture were inadequately addressed. This study aimed to investigate the impacts of soil and water conservation practice on soil moisture in the Debre Mewi and Sholit watersheds, Abbay basin, Ethiopia. We compared soil moisture between the treated (Debre Mewi) and the untreated (Sholit) watersheds with SWCs, based on Sentinel-1A data and the field-measured soil moisture, Leaf Area Index (LAI), and water cloud model (WCM). Field-measurement was based on satellite-synchronized 63 soil moisture samples, systematically collected from the two treatment slope positions, two treatment positions, and two depths. We employed ANOVA to compare samples and discern patterns along space and time. The result indicated that the LAI, a predictor of crop yield, was higher in the SWC treated watershed, demonstrating the potential of conserving moisture for boosting crop production. In addition, the results reveal that the higher soil moisture was recorded on the grasslands of the treated watershed at a depth of 15–30 cm, while the lowest was from croplands and eucalyptus trees at 0–15 cm depth. A higher correlation was observed between the measured and estimated soil moisture across three stages of crop development. The soil moisture estimation using WCM from the Sentinel-1 satellite data gives promising results with good correlation ($R^2$ = 0.69, 0.43 and 0.75, RMSE = 0.16, 2.24 and 0.02, and in Sholit (0.7539, 0.933, and 0.3673 and the RMSEs are 0.17%, 0.02%, and 1.02%) for different dates: August, September, and November 2020, respectively. We conclude that in the face of climate change-induced rainfall variability in tropical countries, predicted to elongate the dry spell during the cropping season, the accurate measurement of soil moistures with the mix of satellite and in-situ data could support rain-fed agriculture planning and assist in fine-tuning the climate adaptation measures at the local and regional scales.

**Keywords:** remote sensing; SWC; Sentinel-1A; soil moisture; water cloud model; LAI; upper Abbay basin; Ethiopia

## 1. Introduction

Land degradation is a severe agricultural problem that poses a massive threat to current and potential food production, affecting the livelihood of Ethiopians [1,2] and other developing countries [3]. The topsoil loss in Ethiopia is estimated at a rate of one billion cubic meters per year [4–6], with annual soil loss rates ranging from 42 to 300 t/ha [7], which is an annual cost of $106 million in monetary terms [8–10]. FAO [11] estimated that

50% of the highlands of Ethiopia were already "significantly eroded" in the mid-1980s, which caused a land productivity decline at the rate of 2.2% per year [12].

Institutionalized SWC programs have been significant since the 1970s [13], partly due to the famines of the 1970s and 1980s with the help of the United Nations Development Program and FAO [14] and myriads of donors. According to a review compiled by [15], SWC related initiatives practiced in Ethiopia include Food-for-Work (FFW), Managing Environmental Resources to Enable Transition (MERET), Productive Safety Net Programs, Community Mobilization through free-labor days, and the National Sustainable Land Management Project (SLMP).

Ethiopia's northern and central highlands have been areas of the dense population for millennia, contributing to severe soil erosion problems, which compelled the government and donors for massive SWC interventions. The centuries-old farming practices depleted the soil organic matter and soil fertility [16]. Haregeweyn et al. [15] identified over 258 studies that dealt with either soil erosion or SWC in Ethiopia, of which 162 focus on the Ethiopian highlands and 112 on the northern Ethiopian highlands. Several studies assessed the effectiveness of SWC interventions [17–22].

A cursory review of SWC studies reveals an overwhelming bias towards quantifying the losses or preserved soil particles from farmlands. For instance, Refs. [11,15] presented the soil loss for Ethiopia at an annual rate of $1.9 \times 10^9$ t; ref. [23] put the Figure as $1.5 \times 10^9$ t; ref. [24] provided a mean annual soil loss ranging from 0 t ha$^{-1}$ year$^{-1}$ in the eastern and southeastern parts to more than 100 t ha$^{-1}$ year$^{-1}$ in the northwestern part of Ethiopia. Those studies focused on the "soil" element of SWC rather than the "water" element of the SWC, partly due to the challenges of acquiring timely soil moisture data from a broader area. The predicted rate of soil erosion would immensely diminish the amount of moisture available for plant uptake due to the reduced volume of pore spaces that could store moisture. In addition, there will be selective removal of finer soil particles that alter soil texture and eventually lower the water holding capacity. On the contrary, the rapidly growing population exerts heightened stress on the farmlands, widening the gap between soil supply and demand and putting food security at risk.

Soil moisture varies horizontally [25] and vertically within the soil solum. In addition to the short supply of rainfall or irrigation, soil moisture variability is controlled by topography [26], soil properties, climate, and vegetation cover [27], as well as ground water table level [28]. Such a scenario calls for an in-depth study of soil moisture behavior, a key ingredient for land use planning, prioritizing land management practices, and running hydrological modeling [29,30].

In a country where most of the population's livelihood and the gross national product rely on rainfed agriculture, which accounts for 40% of the GDP, 80 percent of exports, and an estimated 75 percent of the country's workforce [31], a slight reduction in soil moisture during the cropping season could risk total crop failure. Several studies [32,33] associated dry spells with drought and food insecurity instances, citing the recorded long dry spells in Ethiopia for over ten days in 1972 and 1987 as an example. Those dry spells are most prevalent in Ethiopia's northern, eastern, and southeastern parts [34–36].

According to CMIP5 and CMIP6 simulations [37], future rainfall distribution will be more erratic, and dry spells will be longer. Therefore, the soil moisture, which is to be retained by the SWC structures, needs to be sufficiently explored, especially in the dearth of studies on the soil moisture aspect of SWC and the contribution of soil moisture to crop production and food security.

Because soil moisture is very dynamic in space and time and the plants mainly cover soils, there is a considerable challenge to acquiring timely moisture information. The traditional gravimetric method [38,39] is time-consuming and requires rigorous laboratory analysis. In contrast with gravimetric methods, remote sensing tools provide efficient approaches for soil moisture retrievals at a large scale with high temporal and spatial resolution and low cost [40]. The most widely used land resource satellite data were mainly detected in the optical (passive) spectral range, having very little relevance to detecting and

monitoring soil moisture. Recently, microwave remote sensing has become accessible [41], and Sentinel 1 is one. Due to its stronger frequencies and longer wavelengths, microwaves can penetrate through clouds and vegetation and are highly sensitive to the water in the soil due to the change in the soil's microwave dielectric properties [42]. Thus, microwave remote sensing data have been successfully used to monitor surface parameters over agricultural regions, including surface soil moisture [43]. However, backscatter's inherent complexity makes the backscattering models' inversion very difficult.

As with theoretical, empirical, and semi-empirical models, different models are commonly used to estimate soil moisture [44]. These models are commonly used to separate both the vegetation cover and surface roughness so that soil moisture can be estimated under sparse and dense vegetation cover. A theoretical model such as the integral equation model (EMP) is complicated and requires many parameters. On the other hand, empirical models, such as the Dubois model, are simple to develop; however, there are limitations for applying it to other sites because of its data and site dependence [45]. Recently, the Dubois model has allowed reliable soil moisture estimation. The semi-empirical model is similar to the Oh model and water cloud model (WCM) [46]. It was often used to estimate soil moisture and model the dispersion of vegetation areas for its simplicity [47]. Moreover, the remote sensing inversion techniques also include Neural Network (NN), and change detection (CD) are also widely used in soil moisture estimation [48]. The neural network approach consists of many hidden neurons or nodes that work in parallel to convert data from an input vector to an output vector. For soil moisture retrieval, the neural network is often trained using a synthetic database generated from a SAR backscattered model such as IEM, Oh [49], and WCM models [50].

In areas completely covered with vegetation, the information on soil radiation is masked by the vegetation cover and influences the accuracy of the soil moisture estimation from C-band Synthetic Aperture Radar (SAR) [51]. Therefore, the SAR is excellent in estimating soil moisture in bare and sparse vegetation areas [52]. Several SAR backscatter models separate backscatter contributions from bare land and backscatter from vegetation [53]. The water cloud model (WCM), which can efficiently perform in an inversion scheme to estimate soil moisture and vegetation parameters, is widely used to estimate soil moisture from remote sensing data such as SAR [47]. This semi-empirical model was used to reduce the backscattering effect in soil moisture estimation in the present study. There is an acute dearth of soil moisture-focused SWC studies and the relevance of conserved soil moisture in the face of change because of rainfall distribution leading to longer dry spells and risking the food security of a rapidly growing population. This study aims to examine the amount and distribution of conserved soil moisture for crop production across the temporal and spatial dimensions over land use, soil depth, and topographic variables. Specifically, the study aims: (1) to assess the performance of the WCM in the estimation of soil moisture under vegetative areas, (2) to investigate the effects of land use land cover types on the soil moisture; and (3) to analyze the soil moisture variation based on topographic position and soil depth. The findings of this study furnish the contemporary academic debate on the SWC-climate change-food security nexus in the context of developing countries.

## 2. Materials and Method

### 2.1. Study Area

The study area is located within the Upper Blue Nile Basin, characterized by the degraded highlands of Northwest Ethiopia. We selected two watersheds, viz., Debre Mewi, located between 11°20′10″–11°21′58″ N and 37°24′07″–37°24′55″ E and Sholit, located between 11°22′20″–11°22′28″ N and 37°24′15″–37°24′16″ E (Figure 1). Administratively, the two watersheds are found in the Yilmana Densa Woreda, near Bahir Dar, the capital city of the Amhara Regional State. The watershed is characterized by different topographic conditions, ranging from flat plains to steep areas. After classifying DEM into five FAO slope classes, slope (0–2%), (2–5%), (5–10%), (10–20%) and >20% (Figure 1). The Debre Mewi watershed covers 586.19 ha, with 54% of its area having a slope exceeding 20%

and had practiced SWC for over 20 years. Sholit watershed, on the other hand, covers 293.55 ha and 50% of its area exceeds 20% slopes, with no SWC practiced in it (Figure 2). The altitude of the study watersheds ranges from 2151.63 to 2343.42 m.a.s.l. The dominant soil types in the two watersheds include Eutric Vertisols, Eutric Luvisols, Pellic Vertisols, Eutric Cambisols, Eutric Fluvisols, and Eutric Aquic Vertisols [45].

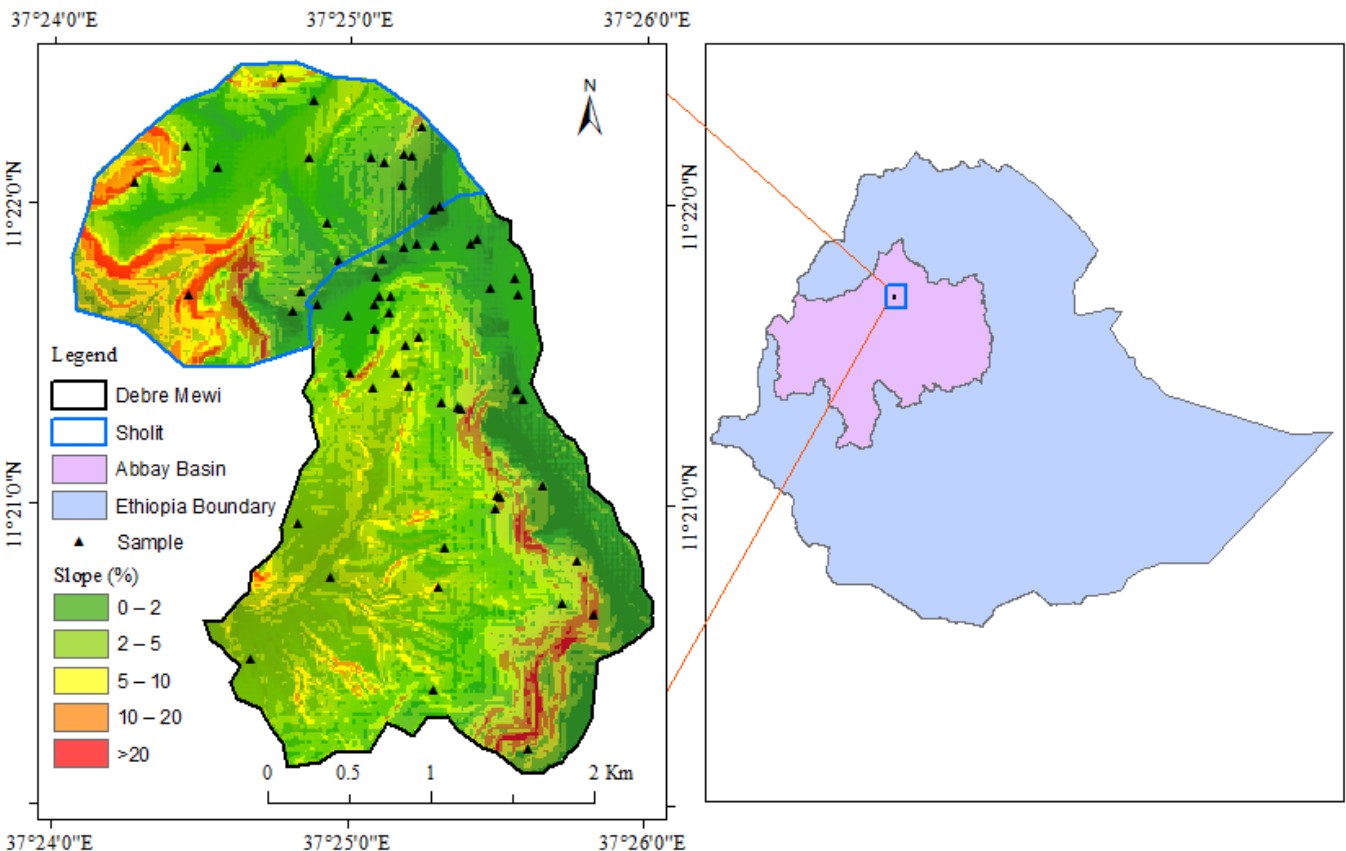

**Figure 1.** Location map of the study area: Sholit (above) and Debre Mewi (lower) watershed and all micro watersheds with corresponding sample location (Source Author developed).

The mean annual rainfall in the study area is 1221 mm, at which more than 80–90% of the rain occurs during the three months of the growing period. During the study period, the average daily temperature was 16.24 °C and 20.25 °C, respectively. Teff (Eragrostis teff), finger millet, and maize dominate the agriculture of the two watersheds [12].

In the Debre Mewi watershed, gully erosion rehabilitation started in the early 1980s, mainly through erecting soil and bunds. Soil bunds dominate the middle and low slope positions, characterized by scarcity of stones, while stone bunds dominate the upper slopes with variable height and width, constructed through mass community mobilizations. Structures are rehabilitated seasonably, and ox plows are used to dispose of excess runoff from farm plots between bunds. Since recently, tree plantations such as *Sesbania sesban* (*Sesbania bispinosa*) and grass species have been integrated into the SWC measure to stabilize bund structures and gully treatments.

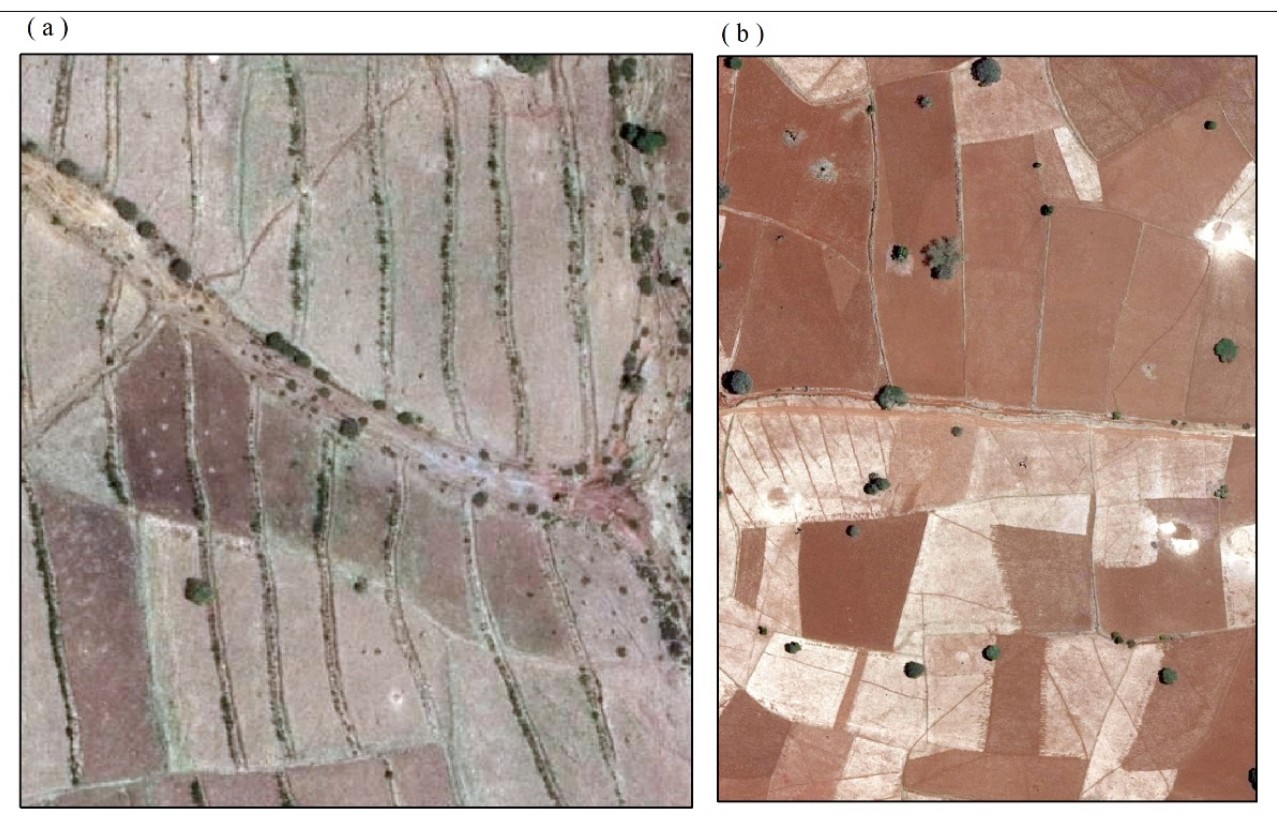

**Figure 2.** Farmlands in the study area: (**a**) treated with stone and soil bunds in Debre Mewi, and (**b**) untreated with SWC structures in Sholit sub-watershed (photo taken: May 2020).

*2.2. Data*

2.2.1. Soil Samples

We identified sample plots during the reconnaissance survey. We employed a purposive sampling technique to collect representative samples from SWC treated (Debre Mewi) and untreated/control (Sholit) watersheds during the actual field data collection. The soil samples collected at 0–15 cm and 15–30 cm soil depths were collected from both watersheds, based on a Randomized Complete Block Design (RCBD) design with split-split plot layout; using two treatments (with and without SWC practices) * 3 slope position (upper, middle, and lower) * 2 soil depths (0–15 cm and 15–30 cm) * 2 replications (upper and lower sides of the bunds). Thirty-two composite samples were collected from both watersheds using the auger. An equivalent number of undisturbed soil samples were used for bulky density and gravimetric soil moisture estimation, collected using core samplers. The soil samples were taken from the site and packed with plastic bags for the laboratory-based soil moisture measurement through the gravimetric method, where the sample soils are weighed, oven-dried at 105 °C for 24 h, and then reweighed. This data is used for the calibration of remotely measured soil moisture data.

2.2.2. Soil Moisture Measurements

We conducted field data collection in 2020, nearly satellite synchronized, at the vegetative, reproductive, and ripening phases of the cropping period in the two watersheds (Table 1). During in-situ soil moisture measurement, we took samples along the slope, land-use type, and availability of SWC in both watersheds. The slope categories include lower (1–15%), middle (15–30%), and upper (30–50%) slopes. We measured the slope angles by a clinometer.

**Table 1.** Date of data acquisition at different growth stages.

| Satellite | Growth Stage | Date of Detection | Field Measurements |
|---|---|---|---|
| Sentinel-1A | Vegetative phase | 19 August 2020 | 9 August 2020 |
| | Reproductive phase | 28 September 2020 | 28 September 2020 |
| | Ripening phase | 22 November 2020 | 25 November 2020 |

Em50 Data Logger was used for the real-time soil moisture measurement, a self-contained data recorder with five channels and designed for use with any ECH$_2$O sensor. We plugged two sensors' tubes into soil profiles at 0–15 cm and 15–30 cm. The EM50 measures volumetric water content via the dielectric constant of the soil using capacitance technology. The sensor uses a 70 MHz frequency, minimizing salinity and textural effects, making it an ideal sensor in agricultural and standard scientific projects [54,55].

We measured total gravimetric soil moisture by calibrating the soil volumetric water content. This instrument was calibrated by Ayehu et al. [45] in northern Ethiopia's Gummara watershed. Measurements on the untreated plot depend on their boundary. We recorded the average real-time soil moisture value using tubes and changed it to (%) (Figure 3). We used a handheld GPS to record the location of sample sites. We executed these procedures three times within the growing season. In total, we acquired 34 measurements from the Debre Mewi and 27 measurements from the Sholit watersheds.

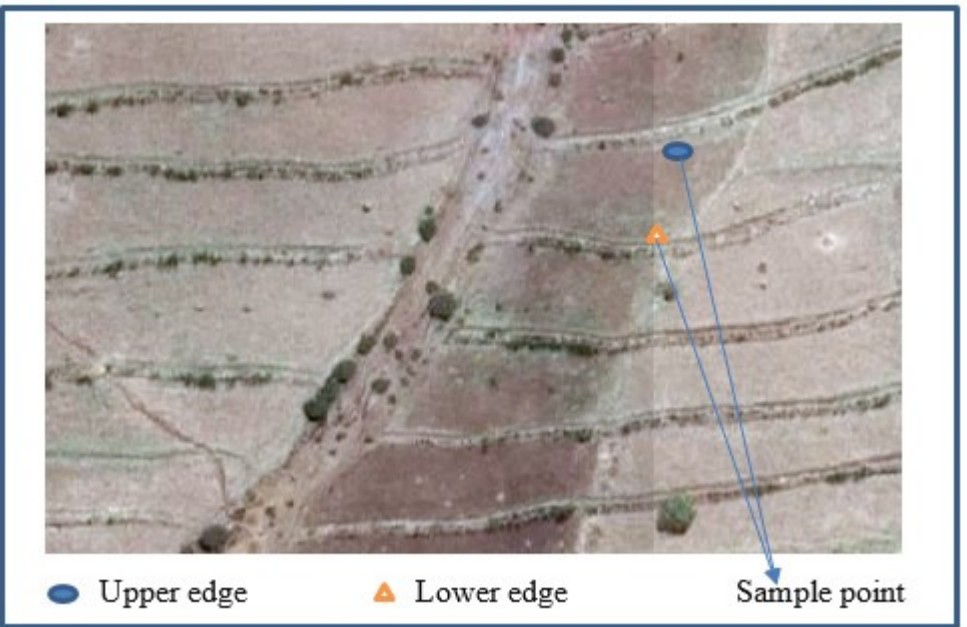

**Figure 3.** Sampling points from plots at Debre Mewi with SWC bunds. While the upper point represents the position below the bund, the lower point represents the above.

2.2.3. Satellite Data

We collected the satellite-based spectral data using Sentinel 1 SAR sensor belonging to the European Space Agency. We downloaded three SAR images at VV/VH polarization between August and November 2020, corresponding to the growing season in the study area from the https://scihub.copernicus.eu/dhus/#/home (accessed on 17 August 2020, 27 September 2020 and 22 November 2020) website. The Sentinel-1A images are all Ground Range Detected (GRD) products in the Interferometry Wide Swath (IW) mode. The range of radar local incidence angles in the study area falls between 300 and 450, and the orbit direction of collecting Sentinel-1A images is ascending. SNAP Toolbox, downloaded from the European Space Agency (ESA), helped preprocess Sentinel-1A images.

We used the Lee method [56] to remove the speckle noise of the images with the window size set to 7 × 7. Similar studies [57,58] have validated this method. We resampled the processed Sentinel-1 images to a 10 m × 10 m resolution and radiometrically calibrated them using a Lee filter for filtering the speckles [44,59]. The geometric correction was done by speckle filtering of images and terrain correction. The radar cross-section was converted from natural values to the backscatter coefficient in dB units.

### 2.2.4. Leaf Area Index (LAI) Measurement

The LAI was used to calibrate the WCM parameters to retrieve soil water content as validated in several studies. LAI data were collected from three dominant crops in both watershed. In this regards from maize (27), teff (21), and millet (15) measurement were collected, respectively. To minimize the error within sampling field, 9 LAI reading was collected from a single plot and averaged to produce one LAI reading for each plot. Because of different factors, there was crop variability within single plots. Sisheber et al. [60] used similar sampling and data collection and recommended that LAI was sampled over 60 × 60 m² elementary sampling units to account for geo-location errors between the field measurements. Within each sampled plot, 10 LAI readings and averaged from multiple positions within the canopy. We measured the LAI of crops in both watersheds from early August to November 2020 using the LAI-2100 plant canopy analyzer placed below the crop canopy (Figure 4). The LAI is measured from the values of canopy transmittance by identifying the attenuation of the radiation as it passes through the canopy [58].

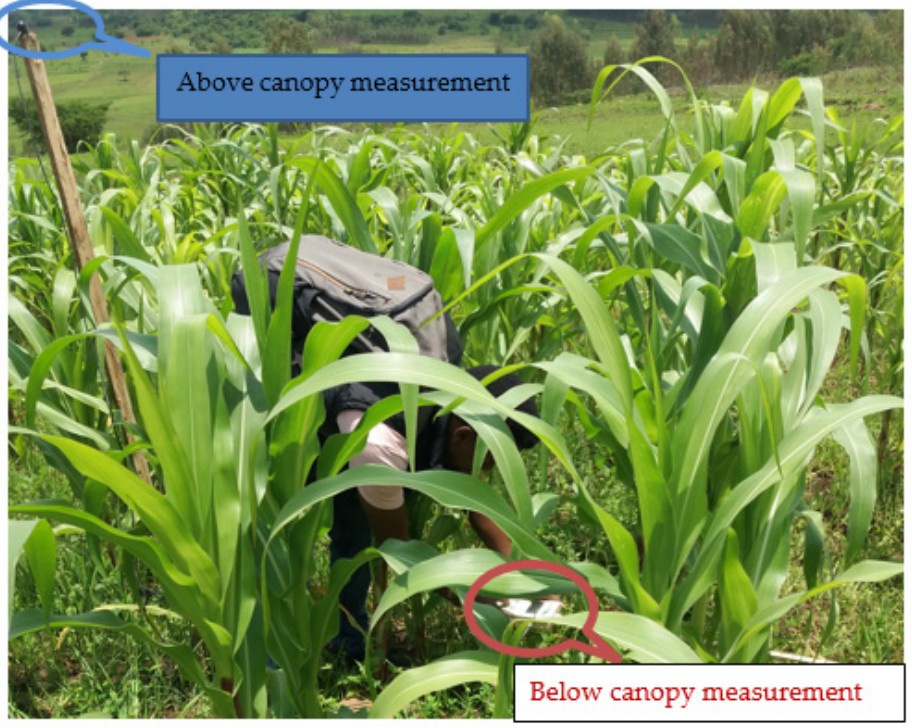

**Figure 4.** LAI measurement by LAI2100 Plant Canopy Analyzer.

We collected LAI samples as the vegetation descriptor of WCM for soil water content retrieval at nine sampling points, measured on teff, maize, and millet in both watersheds. A total of 63 LAI data were recorded from both watersheds in the early mornings under diffuse light conditions, following Heuvelink's recommendation [61,62] to avoid direct sunlight. The satellite synchronized data collection was carried out from 9:00 am to 11:30 am.

### 2.3. Data Analysis

2.3.1. Land Use Classification

We used multi-temporal Landsat 5 Thematic Mapper (TM) images of 1990 and Sentinel 2 of 2020 for mapping LULC types. We used a hybrid classification technique, which combines both unsupervised and supervised classification. Several studies have suggested that hybrid classification produces superior results than unsupervised or supervised alone [63,64]. All images were studied by assigning per-pixel signature and differentiating the watersheds in to 4 major land use classes. The identified land use classes were cultivated land, grazing land, eucalyptus tree, and mixed forest. For each of the predetermined LULC types, the training samples were selected by delineating polygons around the representative place. After the training signature was collected, the maximum likelihood classification algorithm was used for the supervised classification of the image. Normally, these types of image classification that is controlled by the experts by selecting the pixel representing the identified classes [65].

We randomly collected 126 Ground Control Points (GCPs) for the supervised classification of 2020 images. The 1990 LULC classification was performed based on Google Earth images of the corresponding year. A GPS-assisted field survey was conducted to delineate representative training sites for supervised classification of the 2020 satellite image. Furthermore, geo-linking techniques and focus group discussions with local elders were also undertaken, especially for identifying mixed forests. The LULC classification identified: cultivated land, grazing land, mixed forest, and eucalyptus plantation. For the accuracy assessment of LULC classes on the 1990 and 2020 images, we collected reference points from the Google Earth images and compared the classified image with the reference images using an error matrix. We used the Kappa, using Equation (2) [66].

$$\text{OA} = \left(\frac{x}{Y}\right) * 100 \tag{1}$$

where OA is overall accuracy, $x$ is the number of correct values in the diagonals of the matrix, and Y is the total number of values taken as a reference point.

$$K = \frac{N \sum_{i=1}^{r} xii - \sum_{i=1}^{r}((xi+) * (x+i))}{N^2 - \sum_{i=1}^{r}((xii+) * (x+i))} \tag{2}$$

where, $K$ is Kappa coefficient, $r$ is the number of rows in the matrix, $xii$ is the number of observations in row $i$ and column $i$, $xi+$ area the marginal total of row $i$, $x+i$ are the marginal total column $i$, and $N$ is the total number of observations.

2.3.2. The Water Cloud Model (WCM)

WCM is the most widely used vegetation scattering model to simulate soil moisture content throughout cropping. It is a canopy scattering model [67,68], which assumes a uniform and single medium vegetation layer. The WCM does not consider the multiple scattering of electromagnetic waves between the vegetation and soil layers [47]. Since agricultural lands consist of vegetation and bare soils, the signals detected by the satellite image combine the two, making it widely used in soil moisture content retrieval over vegetated areas. WCM assumes that the total radar backscatter ($\sigma^0$) consists of vegetation layer scattering ($\sigma^0_{veg}$) and soil layer scattering ($\sigma^0_{soil}$). Given that the scattering information from the soil layer is affected by the vegetation layer, a two-way vegetation attenuation value ($\tau2$) is used to correct the deviation caused by this effect (Equations (3) and (4)). The scattering from the vegetation layer is described by Equation (5).

$$\sigma^0 = \sigma^0_{veg} + \gamma \sigma^0_{soil} \tag{3}$$

where $\sigma^0_{veg}$ and $\sigma^0_{soil}$ represents the parts of the overall backscatter coefficient $\sigma^0$ from vegetation and soil respectively.

$$\gamma^2 = \exp^{(-2*B*LAI*\sec\theta)} \tag{4}$$

$$\sigma^o_{veg} = A * LAI * \cos\theta \left(1 - \gamma^2\right) \tag{5}$$

where LAI is the canopy descriptors from field measurement [69]. LAI is used for describing the characteristics of vegetation, θ is the radar incidence angle, and A and B are parameters that depend on the canopy type [70].

In the WCM, the scattering from the soil layer is mainly linked to the soil moisture (Sm) and surface roughness, which can be described by a classic linear function of soil moisture (Sm) in Equation (6).

$$\sigma^0_{soil} = C * Sm + D \tag{6}$$

where C and D are parameters depend on soil roughness that characterize the soil. The parameter D is the sensitivity of the signal to soil moisture, and C can be considered a calibration constant [46,71], obtained by fitting the model on experimental datasets.

In Equations (4)–(6), the WCM model has four parameters: A, B, C, and D, which need to be calibrated. The in-situ soil moisture was used for the calibration. Therefore, the general expression of soil moisture (Sm) obtained by the numerical inversion model [53] can be expressed using Equation (7).

$$Sm = \frac{\left\{ \frac{\sigma^0_{vv} - A*LAI*\cos\cos\ \theta(1-\exp\exp\ (-2*B*LAI*\sec\theta))}{\exp(-2*B*LAI*\sec\theta)} \right\} - D}{C} \tag{7}$$

In Equation (7), we used field-measured soil moisture data to represent Sm and LAI obtained from the field. θ is derived from the Sentinel-1 backscattering coefficient and the Sentinel-1 incidence angle, respectively. Of the 63 experimental sampling points, half were used to train WCM and the remaining half to validate the soil moisture estimation accuracy. Thus, we will have 32 equations in the form of Equation (7), where A, B, C, and D are vegetation and soil parameters to be determined (Table 2).

**Table 2.** Calibrated values of WCM parameters.

| Data Acquisition Date | A | B | C | D |
|:---:|:---:|:---:|:---:|:---:|
| August | −6.12 | −1.48 | 0.02 | −5.62 |
| September | −1.28 | −2.62 | 0.0017 | −0.85 |
| November | −22.16 | −3.12 | 0.059 | −6.82 |

### 2.3.3. Statistical Analysis

We used SPSS, version 20, to calculate a one-way and two-way ANOVA test. One-way ANOVA was used to test the significant mean difference of soil moisture between treated and untreated watersheds. Two-way ANOVA was performed to evaluate the influence of land use type, soil depth, and topographic position on soil moisture. Therefore, the statistical analysis considered the SWC practice, control watershed, slope position, and land use type as independent variables and soil moisture as dependent variables. The soil moisture retrieved from Sentinel-1-based WCM was evaluated with in-situ measurements with the help of statistics coefficient of determination ($R^2$), root means square error (RMSE), which signifies the closeness of two independent datasets (estimated vs. measured). Studies [45,72] advised using these statistical analyses to compare the estimated values with measured data. The coefficient of determination ($R^2$) and Root Mean Square Error (RMSE) was used for this study.

## 3. Results and Discussion

### 3.1. Leaf Area Index (LAI)

Based on a total of 63 LAI data measured from nine sampling points of the two watersheds, the result revealed a higher LAI for the SWC treated watershed than the untreated watershed at all stages of the crops (Figure 5), which is consistent with [73], who reported a higher LAI and yield from the SWC treated plots than the untreated plots. For the SWC treated watershed, millet showed an extreme LAI variation, ranging from 0.06 in the vegetative stage to 4.02 in the ripening stage. In contrast, the record for untreated watershed ranged from 0.17 in the millet vegetative stage to 4.1 in the maize ripening stage. Except for millet, the LAI increased along with the crop development: from vegetative to ripening in both SWC treatments, a result that aligns with the findings of [74].

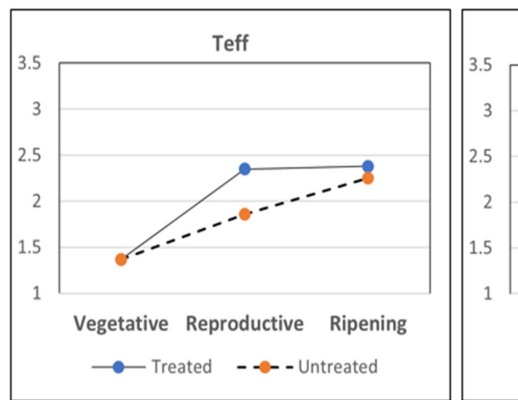
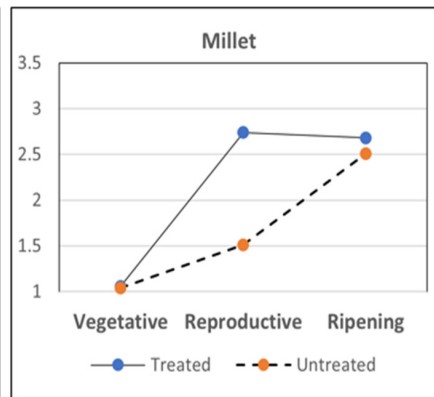
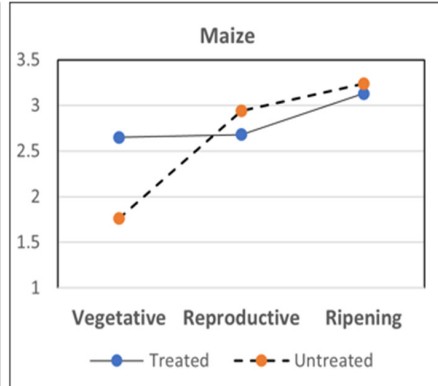

**Figure 5.** Leaf area index (LAI) of teff, millet and maize across growth stages in the SWC treated and untreated watersheds.

LAI was different under various crop crowing stages. LAI was low in the early vegetative and ripening stage observations and peaked by reproductive stage (maize and millet) and teff. The maize and millet LAI readings were higher than teff due to the canopy difference. The maximum LAI observed from three crops at reproductive stages when the crop had accumulated maximum biomass content. The LAI then decreased towards harvest time. According to He et al. [75], the LAI was increased from the transplanting to reproductive phase and the increment rate was significantly slowed after the reproductive phase.

In our study, crop yield was not measured. However, based on the positive linear relationship between LAI and crop yield [76] and the prediction of crop yield as a function of the logarithmic LAI [77], we could safely speculate a higher yield to be obtained from SWC treated watershed than from the untreated. Further, the conserved soil could furnish nutrients to the crops due to higher microorganism activities in the wet soils [78] in the treated watershed.

### 3.2. WCM-Based Soil Moisture Estimation

Figure 6a–c shows the estimated soil moisture using the WCM for the three episodes of the cropping season: 19 August, 28 September, and 22 November 2020. Overall, the average soil moisture varies from 36.05% to 38.74%, with a standard deviation of 1.57, which is relatively higher in August and September at the lower slopes of the two watersheds. Soil moisture varies from 32% to 45% in the Debre Mewi watershed and 20% to 45% in the Sholit watershed.

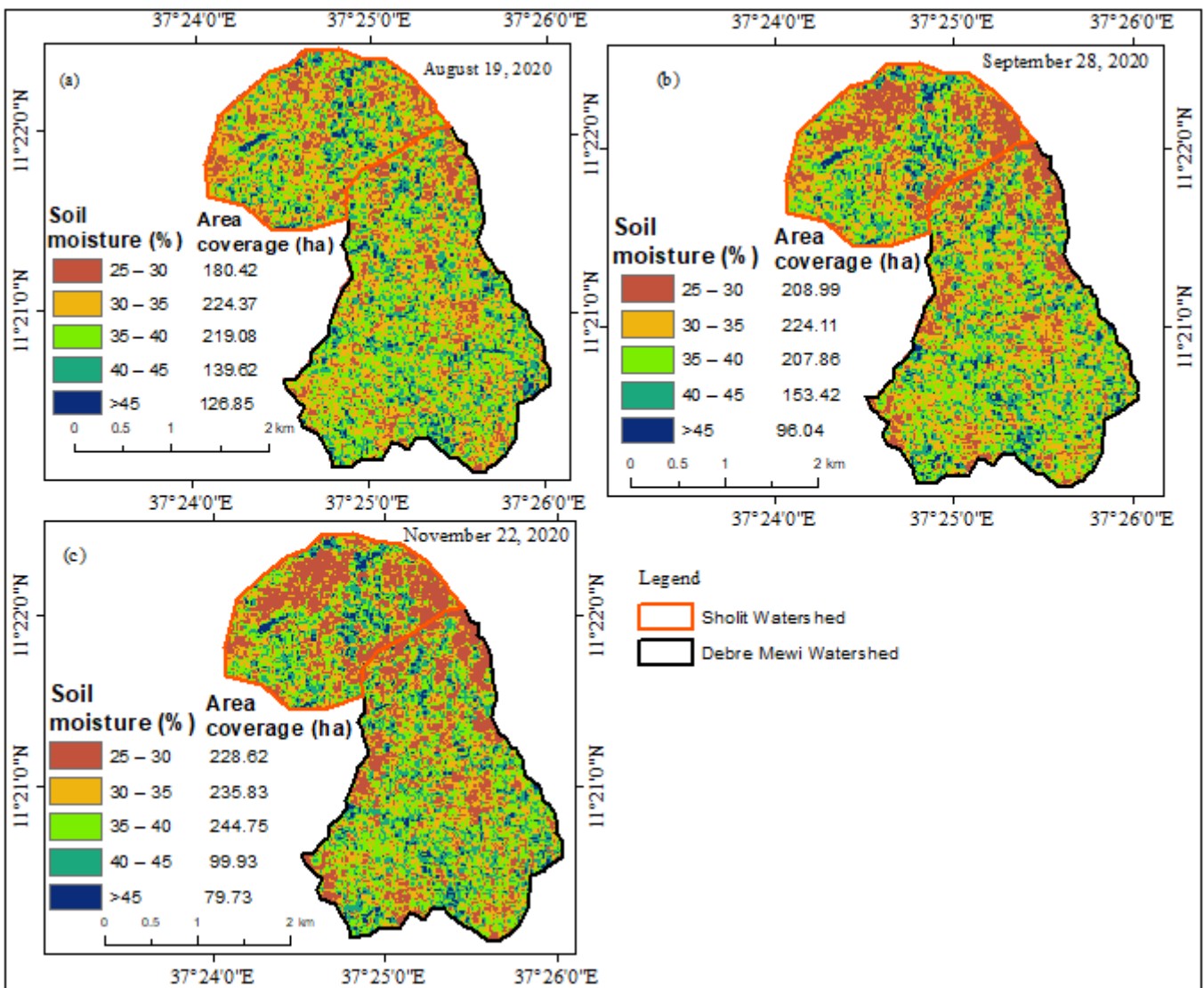

**Figure 6.** Soil moisture class and the area coverage (in ha), derived from WCM (**a**) August 19 (early vegetative phase), (**b**) September 28 (reproductive phase), and (**c**) November 22 (ripening phase).

Figure 6a–c further reveals the soil moisture estimated from WCM, corresponding to the area covered in each class. The maps show that the soil was almost saturated at the early stage of the cropping season. Therefore, the area coverage in the lower class is small, and the higher class covered a large area. From September to November, the area coverage related with soil moisture class, soil moisture constant was decreased from higher class to lower class. In general, the area coverage in the lower class (25–30% and 30–35%) increased from August to November, and in the higher class (40–45% and >45%), the area coverage decreased. Since the rainfall in the study area declined after September, the soil moisture content correspondingly reduced to the lower class and showed increased area coverage expressed in hectares. In addition, we can see from these results under three study seasons that the lower soil moisture class was under cropland.

The estimated soil moisture showed a spatiotemporal pattern. The estimate exceeded 45% in both watersheds in August and September, mainly in the lower parts. After the cessation of the rainy season, the estimate was reduced to less than 10% at the beginning of November. Right after the sowing period, between mid-June and August, the effect of crops on the acquisition of soil moisture signal backscatter is minimal. After the crops cover the soil, from September to mid-October, using WCM becomes challenging.

In our study, a variable effect of vegetation is observed at different growth stages: lower in the ripening stage than the reproductive stage. For that reason, the accuracy of soil moisture estimation was better with less vegetation due to the little hindrance of vegetation. Hence, the accuracy of soil moisture retrieved with the WCM in August was the least among the three periods. The study results confirmed that if the surface roughness property is not properly represented in the retrieval model, the accuracy of the model's output will be adversely affected. Since soil moisture was very sensitive by surface roughness. Therefore the accuracy of the retrieved soil moisture from sentinel -1 using WCM depends on the surface coverage.

Figure 7 shows that the mean soil moisture recorded on the treated watershed was 33.25 ± 1.65 and 32.08 ± 1.21 from treated and untreated farm plots, respectively. Such difference could be due to the reduced slope length of SWC structures that impedes the runoff and enhance soil water holding capacity. In addition, the marked increase in soil moisture under the SWC practices could be attributed to their influence on water storage in the soil profile. This implies that the SWC practices contribute to fending off the surface flow of rainwater and facilitating the infiltration into the subsoil in the treated watersheds of Debre Mewi. In fact, we observed surface water trapped between consecutive bunds for several days in the treated watershed during the rainy season. This phenomenon enhances the infiltration of rainwater and preserves soil moisture for a long time. Further, the SWC minimizes the loss of fine fractions of soil, including humus, which ultimately enhances the water holding capacity of the soils. A higher soil moisture content observed on SWC treated farmlands aligns with the findings of [79,80].

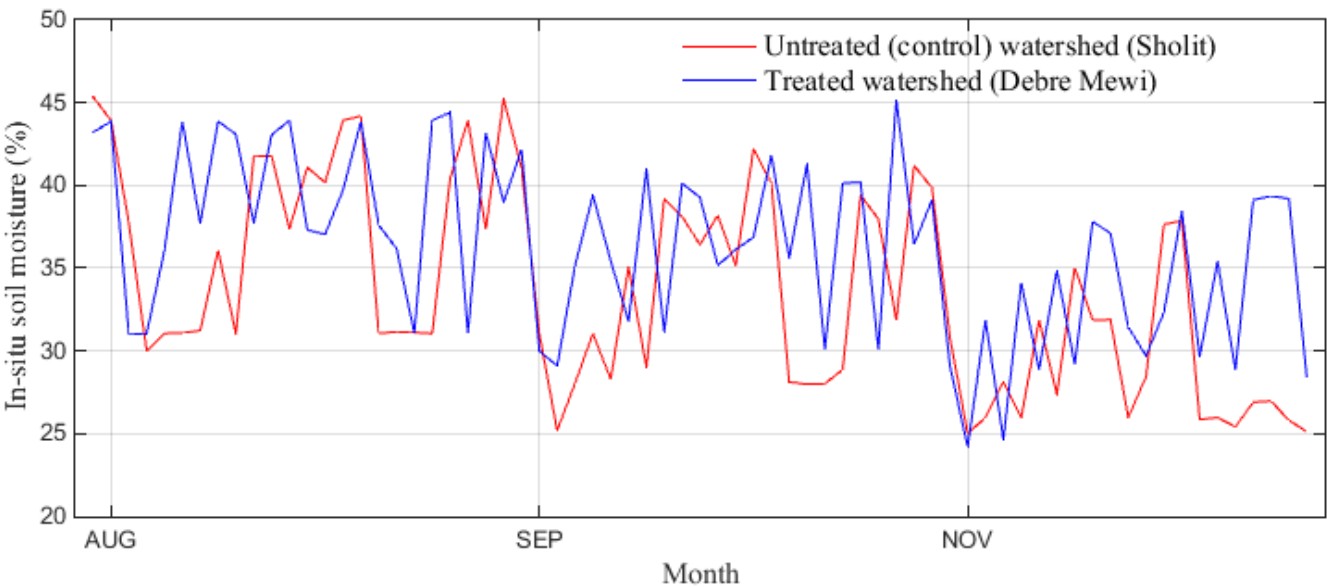

**Figure 7.** Soil moisture variation during the cropping season: August–November 2020.

Figure 7 further shows a different temporal variation between the two watersheds. Immediately after the rainy season, the soil moisture estimates of the Sholit watershed (untreated) showed a quick decline. The treated watershed absorbs and holds water during rainfall events and delivers it to plants during dry spells. Such variation could be very significant, leading to crop failure if the rainfall ceases during grain-filling. Other studies [81–83] confirmed our results by recording higher soil moisture content from treated plots than the untreated plots. The untreated watershed would likely have shallower soil, experience a drier condition, lower soil organic matter, and suffer during dry spells.

### 3.3. Soil Moisture and SWC Practice

In Debre Mewi (treated) watershed, the highest soil moisture content was observed at the upslope and downslope of SWC structure, compared to locations far away from the structures (i.e., midway between the two successive SWC structures). The retained soil moisture adjacent to SWC structures gains a longer time to percolate, thereby contributing to higher soil moisture storage in the soil solum. The higher soil moisture has significant implications for plant uptake close to the structures and groundwater recharge [84].

Table 3 shows the impacts of SWC practice on soil moisture, where the treated watershed showed a statistically significant difference at ($p \leq 0.05$) compared with the untreated watershed. In this regard, the higher mean soil moisture was recorded under treated watershed than the untreated watershed. The result justifies how SWC practices play a vital role in soil moisture content, providing favorable conditions for crop production. This result is congruent with [79,85] and GIZ (2017), who reported increased soil moisture on the SWC treated watershed. Not all SWC interventions demonstrate significant impacts. For instance, [86] argues that croplands with level soil/stone bund and non-terraced do not show remarkable differences for some parameters in Southern Ethiopia.

**Table 3.** Effects of SWC practice on soil moisture in treated and untreated watersheds.

| Study Watersheds | Watershed Status | Soil Moisture % | | | |
|---|---|---|---|---|---|
| | | Mean | Mean Difference | T Value | Significance |
| Debre Mewi | Treated | 34.65 | 1.71 ** | 1.62 | 0.007 |
| Sholit | Untreated | 32.94 | | | |

** shows the mean difference is statistically significant $p \leq 0.05$.

During the data collection in the cropping season, we observed surface water trapped between consecutive bunds lasting for several days in the treated (Debre Mewi) watershed. This phenomenon allows rainwater to infiltrate and store the soil moisture for a long time after the rain stops and reduces the risks of total crop failure due to dry spells.

The added soil moisture due to SWC could boost the soil organic matter [16], ultimately reducing runoff much higher than in the un-treated farmlands. Since clay dominates the soil textural class of the study area, and clayey soils are highly susceptible to erosion [87], the SWC practiced in the treated watershed could significantly increase soil moisture.

### 3.4. Soil Moisture and Topographic Positions

The topographic position-based analysis reveals that the soil moisture varies across a toposequence, which increases the amount from upper to lower topographic positions, and the pattern is confirmed by [85]. The moisture gradient pattern results from enhanced transportation of the finer particles of the soil towards the lower slope positions. According to (Table 4), soil moisture is higher in both watersheds' lower slopes (1–15%). In the middle slope (15–30%), the soil moisture content was declined in both watersheds. The lowest soil moisture was recorded in the upper topographic position. Still, the soil moisture content recorded in all slope categories in the lower slope of a treated watershed was higher than the untreated watershed. The difference could be attributed to the SWC practice that modified the landscape structure [16]. Terefe et al. [79] noted that the height of the stone bunds could reduce the slope gradient by 2.57%.

**Table 4.** Impacts of slope positions on soil moisture in Debre Mewi and Sholit watersheds.

| Slope Position | Watershed Status | | Soil Moisture |
|---|---|---|---|
| Lower (1–15%) | Treated | Mean | 37.24 a |
| | Untreated | Mean | 36.03 b |
| | | *p* values | <0.05 |
| Middle (15–30%) | Treated | Mean | 34.25 a |
| | Untreated | Mean | 32.548 b |
| | | *p* values | <0.05 |
| Upper (30–50%) | Treated | Mean | 30.00 a |
| | Untreated | Mean | 29.12 b |
| | | *p* values | <0.05 |

Note: Under each column followed by letters (a and b) are significantly different from each other at $p \leq 0.05$.

### 3.5. Soil Moisture and Soil Depth

This section presents the results of two assessments: the mean difference of soil moisture along soil depths and the differences of soil moisture across growing stages in the treated and untreated watersheds. The highest soil moisture is observed during the rainy season in August and early September. During the reproductive stage of the crop, the soil moisture in both watersheds showed a similar value (Table 5), which could be due to the complete saturation of soil at this stage. Compared to Sholit (untreated), higher moisture content was observed in the Debre Mewi (treated) watershed during the ripening stage. Our result aligns with [88], where treated plots conserve more water for a long time. In both watersheds, soil moisture is higher in 15–30 cm depth (Table 5), which might be due to the rapid decline of soil moisture on the upper surface layer of the soil, especially during the drier period. The result reported by [89] also showed how the soil moisture content varied with the depth.

**Table 5.** Effects of SWC practice and topographic positions on soil moisture.

| Landscape Position | Vegetative Stage | | Reproductive Stage | | Ripening Stage | |
|---|---|---|---|---|---|---|
| | 0–15 cm | 15–30 cm | 0–15 cm | 15–30 cm | 0–15 cm | 15–30 cm |
| Treated upper | 42.65 | 46.20 | 41.63 | 46.34 | 31.02 | 34.05 |
| Treated lower | 43.93 | 49.01 | 40.21 | 44.36 | 29.96 | 32.34 |
| Untreated upper | 39.55 | 41.03 | 39.71 | 40.23 | 32.09 | 30.36 |
| Untreated lower | 41.95 | 42.36 | 39.00 | 41.89 | 29.32 | 31.74 |

Soil moisture in the subsoil exceeds the topsoil across all situations except for the reversed situation in the upper position of the untreated watershed at the ripening stage. There is an overall declining pattern of soil moisture both for the treated and untreated across the three stages of the cropping season, with no difference between topsoil and subsoil. The only exception is the untreated watershed in the upper positions of the treated and untreated watersheds, where the amount of soil moisture exceeded in the reproductive stage.

When upper and lower topographic positions of treated watersheds are compared, the moisture content gap gets wider between the topsoil and subsoil, with little difference at all stages of vegetation. On the other hand, the untreated watershed's moisture content gap was negligible between the topsoil and subsoil. It implies that the subsoils store very little reserve moisture in the untreated watershed, which is a determinant for crops, mainly when early cessation occurs, which is very common in semidry agro-ecological zones of Ethiopia.

As shown in (Table 4), the soil moisture content significantly varied between upper and lower landscape positions ($p < 0.05$). The lower landscape position showed the highest mean value of soil moisture than the upper landscape positions at all stages of cropping season (Table 5). The result suggests the influence of landscape position on the soil moisture content, especially during the ripening stage of the cropping season.

Although soil moisture availability varies across landscape positions, the overall soil moisture content in the treated watershed showed more moisture than the untreated ones, confirmed by [90], who reported the lowest soil moisture depletion rate in the treated plot compared with untreated plots. The treated watershed (Debre Mewi) showed higher mean soil moisture than the untreated watershed (Sholit), both at 0–15 cm and 15–30 cm depths (Table 5). In the 0–15 cm soil depth, soil moisture content under the condition of the treated watershed was increased by 3.2% and 6.8%, respectively, compared to untreated watersheds. Mekonnen et al. [88] stated that the effect of SWC structures on soil moisture is more important at greater depth, i.e., below the root zone than topsoil, due to little exposure to evapotranspiration than the topsoil. In Debre Mewi (treated) watersheds, the highest soil moisture content was found near SWC structures and declined as one went further away from the SWC structures.

### 3.6. Soil Moisture and Land Use

Figure 8a,b shows the land use/land cover (LULC) maps of the two watersheds in 1990 and 2020, respectively. The dominant LULC types include cultivated land, grazing land, eucalyptus tree, and mixed forest (Table 6). Cultivated land is the most dominant type of land use in both watersheds. In 2020, the area coverage of eucalyptus plantations showed an increasing trend in both watersheds, but the spatial distribution lacks continuity. The spatial representation of LULC types from 1990 to 2020 based on four classes extracted from satellite images with proportionate coverage area is presented (Table 6). In this regard, in 1990, the patterns of LULC as the percentages of the total area studied was dominated by cultivated land covering 68% and 60% of the total studied area in Debre Mewi and Sholit watershed, respectively. In 2020, the patterns of LULC shows that cultivated land was decreased from 68% to 64% in Debre Mewi watersheds, while in the Sholit watershed from 60% to 57%. In contrast to cultivated land, the area coverage of eucalyptus trees was increased from 1990 to 2020.

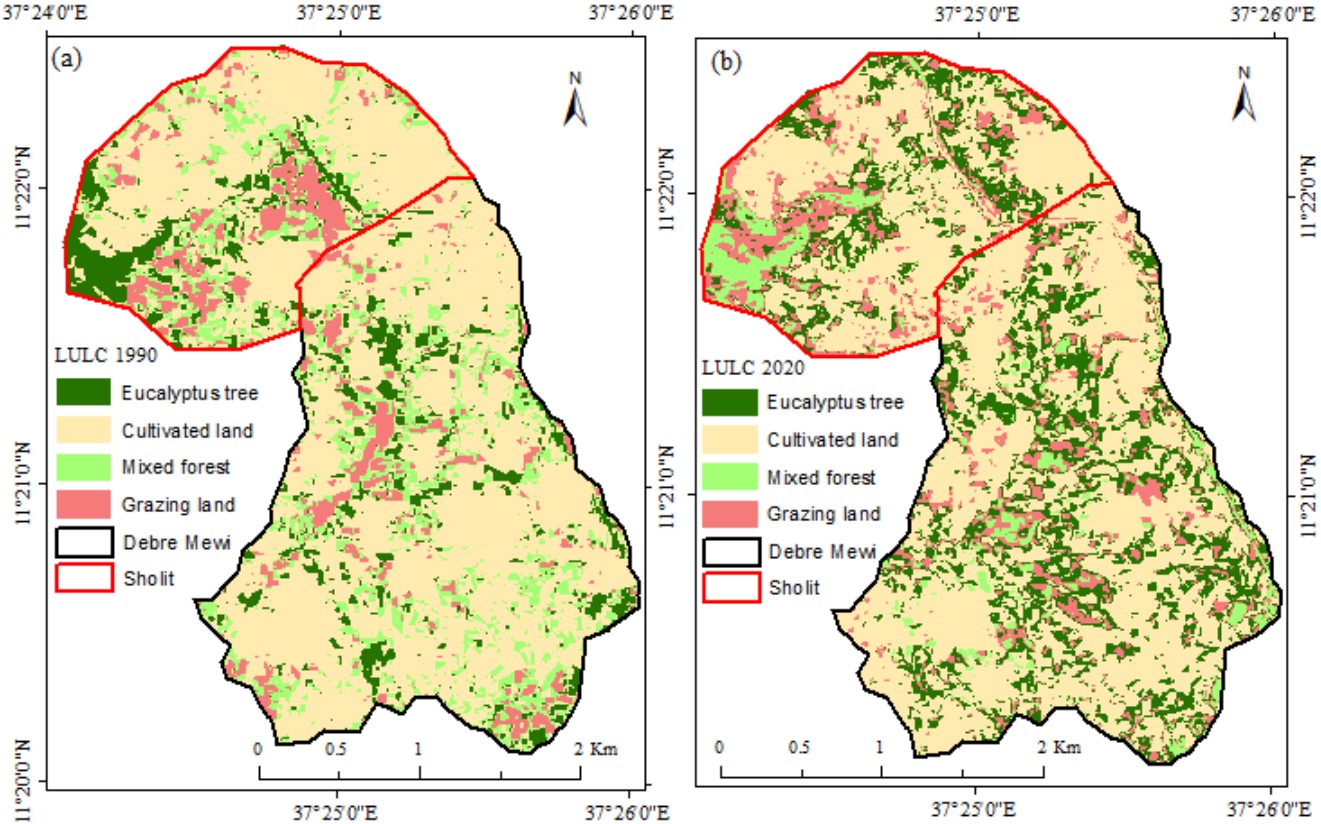

**Figure 8.** LULC maps of Debre Mewi (Bottom) and Sholit (Top) watershed (**a**) in 1990 and (**b**) 2020.

**Table 6.** Land use/land cover change in Debre Mewi and Sholit watersheds over the last three decades (1990 to 2020).

| Land Use Land Cover Classes | Debre Mewi Watershed | | | | Sholit Watershed | | | |
|---|---|---|---|---|---|---|---|---|
| | 1990 | | 2020 | | 1990 | | 2020 | |
| | Area (ha) | Area (%) | Area (ha) | Area (%) | Area (ha) | Area (%) | Area (ha) | Area (%) |
| Cultivated land | 400.59 | 68 | 376.76 | 64 | 176.10 | 60 | 168.41 | 57 |
| Eucalyptus tree | 38.63 | 7 | 125.89 | 22 | 24.27 | 8 | 53.84 | 18 |
| Grazing land | 50.44 | 9 | 55.69 | 10 | 43.16 | 15 | 47.45 | 16 |
| Mixed forest | 95.66 | 16 | 26.96 | 5 | 49.61 | 17 | 23.53 | 8 |

During the big rain period, the soil moisture variation among the LULC classes in the treated watershed was minimal except for the eucalyptus plantation, which conforms with [91]. The farmers in the study area grow eucalyptus trees as the boundaries of farmlands for their economic advantage and runoff and erosion reduction. Refs. [90,92] confirmed that eucalyptus planted around farm boundaries reduces the surface runoff. However, it adversely affects nutrients and moisture contents, ultimately reducing crop yield [92,93].

Inspection of LULC changes in the last three decades (Figure 8) reveals a more clustered and contagious spatial extent of mixed forests in the treated than in an untreated watershed. An expanded but fragmented eucalyptus tree cover is noted in the treated watershed, while a random expansion of eucalyptus is observed in the untreated watershed. In terms of grazing land, one can notice a spatial shifting from the southwest and central part to the northwest area concentration (replacing eucalyptus trees) in the untreated watershed.

In both watersheds, the lowest soil moisture was recorded from lands under eucalyptus plantations than other classes. The difference in soil moisture was not significant during the wet season due to excess rainfall. However, the soil moisture under the eucalyptus trees gets rapidly depleted towards the end of the growing season. The reason might be due to the higher water abstraction by eucalyptus trees. Refs. [90,94,95] also reported the adverse effect of eucalyptus plantation on soil moisture, especially after the rainy season.

The highest soil moisture was recorded in grassland in both watersheds, while the lowest was recorded under the croplands (Table 6), for the fact that grassland intercepts raindrops, reduces surface runoff, and enhances infiltration [96–98]. Likewise, [93] reported no significant difference in a runoff with grassland from eucalyptus-dominated plantations with limited understory vegetation.

As shown in Figure 8, LULC treated with SWC practices showed comparatively higher soil moisture content than LULC classes without SWC with varying amounts of moisture content. For grassland, for instance, it was 38.75% and 37.3% for Debre Mewi and Sholit watershed, respectively, during the vegetative stage. The amount fell to 37.2% and 36.7% for the two watersheds during the reproductive stage. The lowest amount, 25.8% and 22.5% were observed during the ripening stage.

*3.7. Validation of Soil Moisture Estimations*

Figures 9 and 10 validate the water cloud model soil moisture inversion results. The results of 11 August and 10 November validation have best precision among the three dates. Since the interference of crop and vegetation on soil moisture estimate was very little at the beginning and end of the crop growing period, making the WCM-based estimation was better (Figure 9). It means that in the case of less vegetation, the disturbance of vegetation is less, and the retrieval accuracy of soil moisture based on WCM is higher and more accurate with measured soil moisture during (August and November) in both watersheds. Zribi et al. [99] proved that the effects of the vegetation during the vegetative and harvest time was less. However, during September, all crop in the study area was in maximum growth stages, and the effects of surface coverage were very high. Therefore, the linear relationship at this stage (September) was less in both watersheds (Figures 9 and 10). On

the other hand, the estimate in September showed a weak correlation with the ground measurement (Figures 9 and 10) due to the complete soil coverage by crops at the reproductive stage of the cropping season. The relatively more accurate estimate for November than the other crop growth stages of teff and maize that are already harvested, thus generating a small, backscattered signal to complicate the Sentinel-1 imagery. As a result, it showed higher accuracy than those measured on the field; the $R^2$ and RMSEs are 0.75 and 0.174%, respectively.

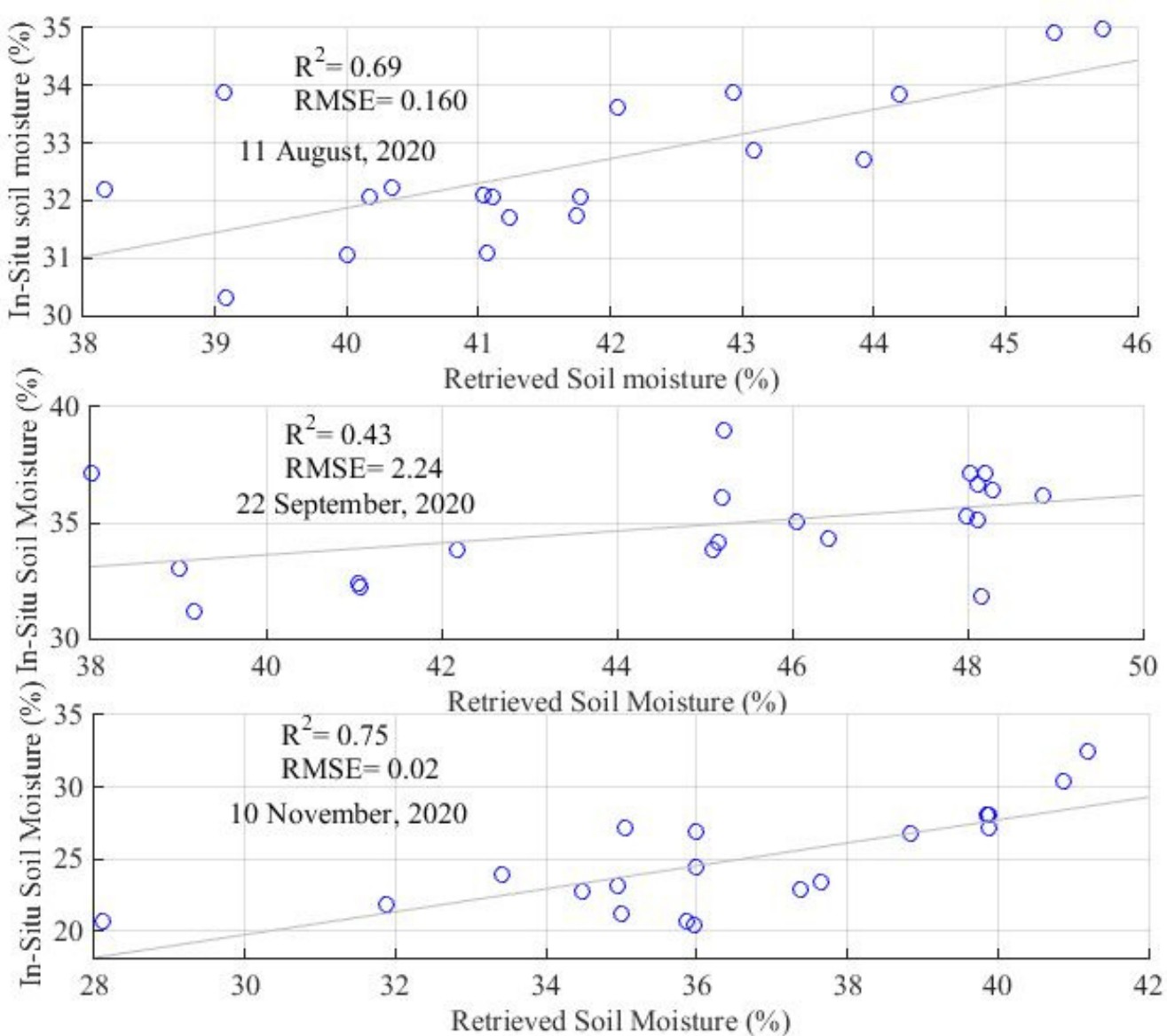

**Figure 9.** Correlation of in-situ soil moisture and WCM-based soil moisture estimate during the cropping season in Debre Mewi (SWC treated) watershed.

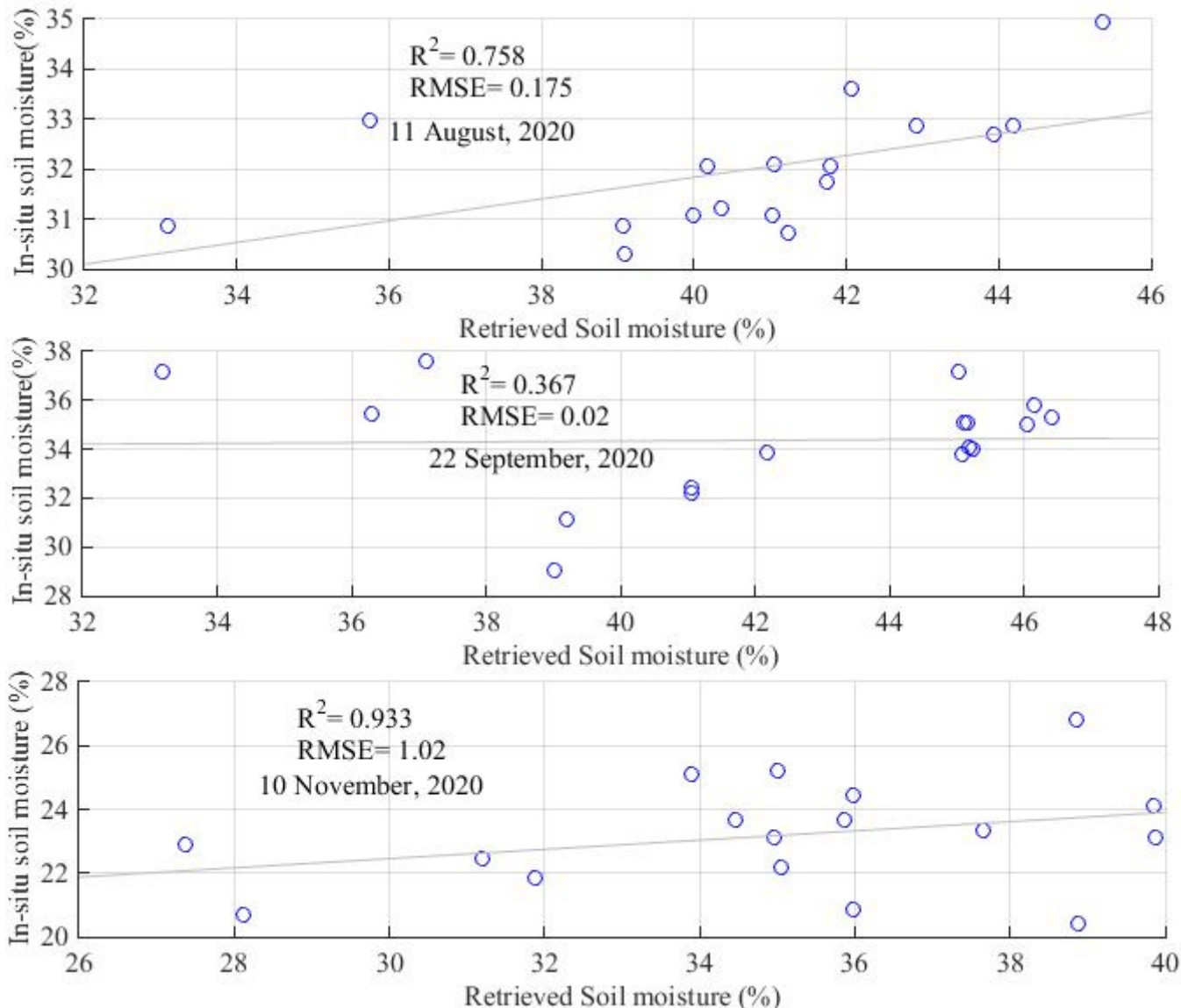

**Figure 10.** Correlation of in-situ soil moisture and WCM-based soil moisture estimate during the cropping season in Sholit (untreated) watershed.

In addition, we obtained different $R^2$ and RMSEs values between observed and estimated soil moisture for the two (treated and untreated) watersheds. In this regard, for August and November 2020, the Sholit watershed shows higher accuracy than that at the Debre Mewi, with the $R^2$ values are (0.75%, 0.93%) and (0.69%, 0.75%) respectively, and the RMSEs are (0.175%, 1.02%) and (0.160%, 0.02%) respectively. This is the reason that grazing land was covered more present when compared with the Debre Mewi watershed so that the effects of the grazing land on the backscatter signal is less than other vegetation coverage. On the other hand, the linear relationship between treated and untreated watersheds during September was also different, with $R^2$ of 0.43% and 0.36%, respectively. The primary reason was the farmers in the Sholit watershed usually cultivated more millet crops than the Debre Mewi watershed. Therefore, the crop calendar between the selected crops was different, by its nature, millet crop needs longtime greens period when compared with teff and maize. This may lead to large errors between the collected ground soil moisture and retrieved soil moisture at the reproductive stage (September), resulting in the soil moisture retrieval results at the Sholit watershed having lower $R^2$ compared to the Debre Mewi watershed.

Different studies [45] support this result. The factors include crop variety [71], different crop growth status [78], and a variable field environment across different areas [41]. In fact, crop types in the treated watershed are dominated by different land use types, such as eucalyptus tree crops (teff, millet, and maize). In this regard, the crop is seasonal, but eucalyptus tree is not. Ground measured soil moisture between treated and untreated watershed is obviously different in August, September, and November. Soil moisture retrieved results at Sholit (untreated) watershed generally have relatively higher $R^2$ values and lower RMSEs than those at Debre Mewi (treated) watershed

Figures 9 and 10 also showed higher soil moisture estimated from the Debre Mewi (treated) watershed expressed in higher $R^2$ values in September. The vegetation coverage affects the estimation and lowers RMSE in Debre Mewi than the estimate from the Sholit (untreated) watershed. The overall accuracy of the LULC classification for the Debre Mewi and Sholit watershed was 88% and 87%, respectively, with kappa coefficients of 0.82 for Debre Mewi and 0.86 for Sholit in 2020. According to [100], Kappa values between 0.70 and 0.85 are a good indicator of the classified image representing the ground truths.

*3.8. Implications of Conserved Soil Moisture for Future Food Security*

Soil moisture is a vital component of water management, rendering a significant challenge to rainfed agriculture. Improved soil moisture retention by SWC practices is consequential when viewed from local and regional biophysical and human variables. Three interrelated scenarios underline the challenges of soil moisture in developing countries. First, the topsoil is removed at an alarming rate, an estimated 75 billion t of soil eroded every year from arable lands worldwide [101], effectively debilitating land carrying capacity. Second, climate change is predicted to pose a growing threat to agricultural productivity from two angles: on the one hand, the atmospheric temperature would surge [102], increasing the evapotranspiration rate and crops' water demands [103]; on the other hand, the rainfall is simulated to become more erratic, creating longer dry spells [37]. Third, the population in developing countries is predominantly agrarian and growing at an alarming pace [104], exerting unsustainable stress on fragile agricultural lands [105]. The composite impacts of soil erosion, climate change, and demographic pressure will likely intensify the scarcity of soil moisture from farm plots and call for research on soil moisture-focused SMC measures in the diverse agro-ecological milieu.

**4. Conclusions**

Soil moisture conserved by SWC measures plays a pivotal role in rain-fed agriculture. However, previous studies focused on the soil erosion of the SWC, expressing the magnitude of the problem in tons per hectare, with little emphasis placed on the moisture component of SWC measures. In this study, we investigated the impacts of SWC practice on soil moisture in treated and untreated watersheds. To analyze these Sentinel-1 data by combing field-measured soil moisture, leaf area index, and water cloud model were used.

Our result showed a higher LAI for the treated watershed than the untreated watershed at all stages of the crops. Treated watershed brought a significant improvement in soil moisture than adjacent untreated watershed. In this regards, treated watershed shows the highest mean soil moisture after rain season. During the vegetative and reproductive stages, the soil moisture content at 0–15 cm shows insignificant differences, while at the ripening stage, the treated watershed shows higher soil moisture content at 15–30 cm depth. Soil moisture varies under different land use types.

The results also reveal that the topographic effect directly bears the soil moisture variability along a toposequence in both watersheds, where the SWC structures have enabled higher moisture content reserve even in the upper topographic positions, potentially expanding the effective croppable area when compared to non-conserved watersheds. In this study, we found that the synergy of ground-based soil moisture measurements, satellite-based moisture estimates, and data analysis helped disclose how the SWC treated watersheds which excelled in their amount of soil moisture translated to a higher LAI.

**Author Contributions:** B.B.D.: Conceptualization, Methodology, Software, Investigation, Formal analysis, Writing—original draft. D.K.W.: Conceptualization, Methodology, Investigation, Writing, review & editing, Supervision. D.A.M.: Conceptualization, Methodology, review and editing, Supervision. D.T.M.: providing field mesurment instrument and Supervision. All authors have read and agreed to the published version of the manuscript.

**Funding:** This research received no external funding.

**Institutional Review Board Statement:** Not applicable.

**Data Availability Statement:** The datasets used during the current study are available from the corresponding author on reasonable request.

**Acknowledgments:** This research was funded by the Ministry of Science and Higher Education of Ethiopia (MoSHE). The authors would like to thank the NASA Land Processes Distributed Active Archive Center (LP DAAC) and US Geological Survey (USGS) for sharing Landsat data. We also thanks the Copernicus Open Access Hub, provides complete, free and open access to Sentinel-1 products, and allow to download Sentinel application platform (SNAP) freely.

**Conflicts of Interest:** The authors declare no conflict of interest.

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
