# Peer review of "Impacts of Soil and Water Conservation Practice on Soil Moisture in Debre Mewi and Sholit Watersheds, Abbay Basin, Ethiopia"

_agriculture, doi:10.3390/agriculture12030417_

Round 1

Reviewer 1 Report

Review

In the manuscript, the authors investigate the impacts of soil and water conservation on soil moisture in Debre Mewi and Sholit watersheds, Abbay basin, Ethiopia, based on the Sentinel-1 data by combing field-measured soil moisture, leaf area index (LAI), and water cloud model (WCM). In general, the authors deal with a hot topic, i.e., soil moisture. However, the manuscript has some drawbacks. I have some major concerns as follows.

Major comment:

1) The abstract is not concise enough. The authors are suggested to reorganize it to better insight the contributions of this manuscript.

2) In the introduction, the review of remote sensing inversion methods for soil moisture should be enhance.

3) In the method, the introduction of classification and statistical analysis is too simple. The equation editing is irregular.

4) In the experiment, the analysis about the results is not deep enough.

5) From Fig. 9 and Fig. 10, part of the linear relationship is not strong, please explain the reasons.

6) The conclusion is too long and not proper. The authors are suggested to reorganized it to emphasize the important findings.

Author Response

Dear Reviewer

thank you very much for your time to comment our manuscript. We take time to review our manuscript depending on your comment and we upload the revised manuscript and response of your comment.

thank again!

best regards

Bekele Bedada

Reviewer 2 Report

The manuscript “Impacts of Soil and water conservation practice on Soil Moisture in  Debre Mewi and Sholit watersheds, Abbay basin, Ethiopia” compared soil moisture between the treated (Debre Mewi) and the untreated (Sholit) watersheds with SWCs,based on  Sentinel-1A data and the field-measured soil moisture, Leaf Area Index (LAI), and water cloud model (WCM). There are some issues to be discussed and improved.

  1. The abstract is too long and too many details are reported. Please, summarize it with only the main results, pointing out the novelty of the study.
  2. It is better to give topographic or geomorphic maps of Debre Mewi and Sholit watersheds in order to understand the impacts of water and soil conservation practice on Soil Moisture.
  3. Please clearly state the object of this paper in the introduction section.
  4. Some of the methods used in the manuscript are not fully explained. How does 63 LAI Data distribute in different crops and their growing seasons?
  5. What is the land use distribution corresponding to "Figure 6. Soil Moisture class and the area coverage"?

Author Response

Dear Reviewer

we would like to thank you for the time you spend to review our manuscript. your valuable comment and suggestion helped us to improve our manuscript. we upload the response of comment.

Thank you again!

best regards

Bekele Bedada

Reviewer 3 Report

"Impacts of Soil and water conservation practice on Soil Moisture in Debre Mewi and Sholit watersheds, Abbay basin, Ethiopia" agriculture-1608750

This manuscript evaluates the effect of soil conservation practices on soil moisture retrieved by explanatory data. The article seems interesting and suitable for the Journal but needs some improvement. However, some comments need to be addressed before the next steps.

1. Citations are not in the format requested by the Journal;

2-I am not a native English speaker, but the quality of English in this manuscript needs revision.

Furthermore, a few comments are indicated in the attached file.

Author Response

Dear Reviewer

Thank you very much for your valuable and specific comment. As you comment on the manuscript by track change we made change direct in the main manuscript. For the spesific comment we respond and upload here

Thank you again

Best Regards

Bekele Bedada

Round 2

Reviewer 1 Report

The authors deal well with my comment. It is ok for me.

Reviewer 2 Report

The revised version of the paper has met the publication requirements, and it is agreed to be published.

Reviewer 3 Report

The authors accurately the previous comments, and the manuscript improved accordingly. I don't have any comments. It can be published in Agriculture journal. Congratulations to the authors.